# Presentation of Congenital Portosystemic Shunts in Children

**DOI:** 10.3390/children9020243

**Published:** 2022-02-11

**Authors:** Atessa Bahadori, Beatrice Kuhlmann, Dominique Debray, Stephanie Franchi-Abella, Julie Wacker, Maurice Beghetti, Barbara E. Wildhaber, Valérie Anne McLin

**Affiliations:** 1Pediatric Specialties Division, Department of Pediatrics, Gynecology, and Obstetrics, University Hospitals Geneva (HUG), University of Geneva, 1211 Geneva, Switzerland; julie.wacker@hcuge.ch (J.W.); maurice.beghetti@hcuge.ch (M.B.); 2Pediatric Endocrinology, Cantonal Hospital Aarau, 5001 Aarau, Switzerland; beatrice.kuhlmann@ksa.ch; 3Pediatric Liver Unit, Necker Hospital, APHP, Paris Centre University, 75015 Paris, France; dominique.debray@aphp.fr; 4Pediatric Radiology, Paris-Saclay University, Hôpital Bicêtre, Hôpitaux Paris-Saclay APHP, 94270 Paris, France; stephanie.franchi@aphp.fr; 5Pulmonary Hypertension Program, University Hospitals Geneva (HUG), University of Geneva, 1211 Geneva, Switzerland; 6Pediatric Cardiology Unit, Department of Pediatrics, Gynecology, and Obstetrics, University Hospitals Geneva (HUG), University of Geneva, 1211 Geneva, Switzerland; 7Swiss Pediatric Liver Center, University Hospitals Geneva (HUG), University of Geneva, 1211 Geneva, Switzerland; barbara.wildhaber@hcuge.ch (B.E.W.); valerie.mclin@hcuge.ch (V.A.M.); 8Child and Adolescent Surgery Division, Department of Pediatrics, Gynecology, and Obstetrics, University Hospitals Geneva (HUG), University of Geneva, 1211 Geneva, Switzerland; 9Pediatric Gastroenterology, Hepatology and Nutrition Unit, Department of Pediatrics, Gynecology, and Obstetrics, University Hospitals Geneva (HUG), University of Geneva, 1211 Geneva, Switzerland

**Keywords:** congenital portosystemic shunt, Abernethy malformation, tall stature, liver nodules, hepatopulmonary syndrome, pulmonary arterial hypertension

## Abstract

Background: Congenital portosystemic shunts (CPSS) are rare vascular anomalies resulting in communications between the portal venous system and the systemic venous circulation, affecting an estimated 30,000 to 50,000 live births. CPSS can present at any age as a multi-system disease of variable severity mimicking both common and rare pediatric conditions. Case presentations: Case A: A vascular malformation was identified in the liver of a 10-year-old girl with tall stature, advanced somatic maturation, insulin resistance with hyperinsulinemia, hyperandrogenemia and transient hematuria. Work-up also suggested elevated pulmonary pressures. Case B: A young girl with trisomy 8 mosaicism with a history of neonatal hypoglycemia, transient neonatal cholestasis and tall stature presented newly increased aminotransferase levels at 6 years of age. Case C: A 3-year-old boy with speech delay, tall stature and abdominal pain underwent abdominal ultrasound (US) showing multiple liver nodules, diagnosed as liver hemangiomas by hepatic magnetic resonance imaging (MRI). Management and outcome: After identification of a venous malformation on liver Doppler US, all three patients were referred to a specialized liver center for further work-up within 12 to 18 months from diagnosis. Angio-computed tomography (CT) scan confirmed the presence of either an intrahepatic or extrahepatic CPSS with multiples liver nodules. All three had a hyperintense signal in the globus pallidus on T1 weighted cerebral MRI. Right heart catheterization confirmed pulmonary hypertension in cases A and C. Shunts were closed either using an endovascular or surgical approach. Liver nodules were either surgically removed if there was a risk of malignant degeneration or closely monitored by serial imaging when benign. Conclusion: These cases illustrate most of the common chief complaints and manifestations of CPSS. Liver Doppler US is the key to diagnosis. Considering portosystemic shunts in the diagnostic work-up of a patient with unexplained endocrine, liver, gastro-intestinal, cardiovascular, hematological, renal or neurocognitive disorder is important as prompt referral to a specialized center may significantly impact patient outcome.

## 1. Introduction

Congenital portosystemic shunts (CPSS) are rare anatomic vascular anomalies resulting in communications between the portal venous system and the systemic venous circulation, affecting an estimated 30,000 to 50,000 live births [1,2]. They are accepted to arise from incomplete vascular remodeling between the embryonic and fetal hepatic and perihepatic circulations in the first 4 to 6 weeks of gestation [3]. These vascular malformations are generally low-pressure systems that may vary in size and number and may exist inside or outside of the liver diverting portal blood flow with varying degrees to the systemic circulation [3,4,5]. CPSS differ from portal hypertension-acquired intra- or extrahepatic portosystemic shunts in that they are low-pressure systems unrelated to portal hypertension.

Extrahepatic CPSS were previously known as Abernethy malformations [6]. Today, CPSS are typically characterized into intrahepatic (IH) and extrahepatic (EH) (Figure 1). This distinction is relevant because extrahepatic shunts rarely close spontaneously [7,8], while intrahepatic shunts seem more prone to do so in early life, decreasing their clinical significance [1,8,9,10,11,12,13,14]. Furthermore, persistent patent ductus venosus (PDV), although considered as an IH shunt, is unlikely to close spontaneously after 1–3 months of age, and therefore is often included in the category of EH CPSS [3].

Both types of CPSS can go unnoticed for a long time, and present at any age masquerading as one of several conditions, highlighting the multiple functions of the liver. By way of example, presentations include neonatal cholestasis [3,5], liver tumors [15,16,17], hepatopulmonary syndrome (HPS), pulmonary arterial hypertension (PAH), high-output cardiac failure [4,5,10,18], hyperinsulinemic hypoglycemia [19], hyperammonemia, hyperandrogenism, precocious puberty, tall stature [3,20,21,22], amenorrhea, hypothyroidism, macrohematuria [23,24] and neurocognitive disorders [1,25,26]. In a recent observational study of 66 adult patients, the cumulative incidence of at least one major EH CPSS manifestation (hepatic encephalopathy, PAH, HPS, hepatocellular carcinoma or adenoma) at 20, 30 and 40 years of age was 35%, 45% and 58%, respectively [9].

Early diagnosis by means of liver Doppler US allows for prompt management of potentially life-threatening manifestations, ultimately improving the outcome in these patients. Therefore, the purpose of the present article is to expand on three cases with variable presentations to discuss the severe clinical consequences of CPSS. We will focus on clinical presentation and initial diagnosis, with a brief overview of management for the general pediatrician, in order to increase awareness in the pediatric community and decrease diagnostic delay. 

## 2. Case Presentation

Table 1 summarizes three cases of CPSS.

### 2.1. Case A

A nine-year-old girl was referred to an endocrinologist for tall stature and advanced pubertal maturation (Tanner 3). Initial work-up revealed insulin resistance with hyperinsulinemia, acanthosis nigricans, hyperandrogenemia and pubertal activation of the pituitary–ovarian axis without menarche. Bone age was accelerated by five years. The following were also identified over the course of the work-up: intermittent hyperammonemia, learning difficulties and fatigue with hyperintense globus pallidus on T1 weighted cerebral magnetic resonance imaging (MRI) and intermittent abdominal pain with macrohematuria. Echocardiogram was suggestive of elevated pulmonary pressures. Ultimately, a vascular malformation was suspected on abdominal US. She was referred to a specialized liver and heart center at age eleven for suspicion of PAH and further work-up.

Angio-computed tomography (CT) scan confirmed the presence of an IH CPSS between the left portal vein and the left hepatic vein mimicking the anatomy of a persistent ductus venosus, with two hepatic nodules in segments VII and VIII. Angiography with balloon occlusion of the CPSS confirmed the presence of a single shunt with a normally formed, albeit hypoplastic right hepatic portal vein. Right heart catheterization confirmed moderate PAH (mean pulmonary arterial pressure (mPAP): 40 mmHg; pulmonary vascular resistance index (PVRi): 7 WUm2). Histology revealed hepatocellular adenomas with nuclear beta-catenin expression, and a somatic mutation in exon 3 of the Catenin Beta 1 gene (*CTNNB1)*. PAH was treated using upfront combined therapy by an endothelin receptor antagonist and a phosphodiesterase type 5 inhibitor. Three months later, the CPSS was successfully closed using an endovascular Amplatzer plug. Finally, liver nodules were surgically resected after four months of normal portal flow. At the 9-month follow-up, pulmonary arterial pressures had almost normalized.

### 2.2. Case B

A six-year-old girl, born late preterm with multiple malformations in the setting of trisomy 8 mosaicism, had a history of neonatal hypoglycemia and transient neonatal cholestasis. She was followed yearly for liver enzyme monitoring and accelerated linear growth of unknown etiology. At age 6, she was referred to a specialized liver center for newly increased aminotransferase levels. Outpatient abdominal US was initially described as unremarkable. She was referred again to a pediatric radiologist the following year who identified a heterogenous, hyperarterialized liver parenchyma with no identifiable main portal vein. Angio-CT scan confirmed the absence of the main portal vein with a shunt from the spleno–mesenteric confluence to the inferior vena cava, and a hypodense liver nodule in segment VI, compatible with a focal nodular hyperplasia (FNH) on MRI. Brain MRI performed two years earlier, retrospectively showed hyperintense signals on T1 weighted images in the globus pallidus. Echocardiogram was unremarkable. Exploratory laparoscopy showed no signs of liver fibrosis but revealed a hypoplastic portal vein upon clamping of the shunt. The extrahepatic shunt was partially banded owing to poor intestinal perfusion upon clamping of the entire vessel. Follow-up 6 months post-banding showed thrombosis in the residual shunt despite anticoagulation. In addition, the thrombosis extended to the superior mesenteric vein and was associated with thrombocytopenia, splenomegaly and grade I esophageal varices, compatible with new-onset mild portal hypertension. However, the left portal vein was permeable, and the liver nodule seemed to be regressing on imaging, with normal histology found on the follow-up biopsy.

### 2.3. Case C

A three-year-old boy with tall stature (>97 percentile) and speech delay presented with abdominal pain for which he underwent abdominal US revealing multiple liver nodules (Figure 2). Hepatic MRI showed arterial enhancement of nodules that were mis-interpreted as hemangiomas. After 18 months, at the age of five, he underwent brain MRI for the work-up of tall stature and speech delay, which showed a hyperintense signal on T1 weighted-imaging in the globus pallidus. Repeat US and re-review of previous imaging confirmed the presence of CPSS between the main portal vein and the inferior vena cava. Basic work-up revealed fasting and post-prandial hyperammonemia, and elevated aminotransferase levels. In addition, echocardiogram was suggestive of PAH, which was confirmed by right heart catheterization (mPAP 37 mmHg, PVRi 4.5 WUm2). The portal venogram with occlusion test confirmed a single shunt, with patent, albeit small intrahepatic portal veins. Liver biopsy was consistent with portal deprivation and hyperarterialization. Liver nodule histology was remarkable for regenerative nodular hyperplasia and a hepatocellular lesion with both FNH-like characteristics and beta-catenin nuclear expression. Surgical closure of the shunt, without previous treatment of the PAH, showed expansion of intrahepatic portal veins, and later on the disappearance of liver nodules, neurological improvement and resolution of PAH, 2 years after closure.

## 3. Development of CPSS

CPSS arise from incomplete vascular remodeling between the symmetric embryonic and asymmetric fetal hepatic and perihepatic circulations in the first 4 to 6 weeks of gestation [27,28] resulting in any of a number of forms of CPSS (Figure 1) [1].

## 4. Typical Clinical Presentations of CPSS

CPSS may present with one or more concurrent sign or symptom. Therefore, it is important to perform a thorough work-up when CPSS are identified.

### 4.1. Incidental

CPSS often have been reported as incidental findings. In one series, as many as 27% of IH and 59% of EH cases were identified on abdominal and/or liver imaging performed for another indication [13]. IH shunts are more likely to be asymptomatic when detected and are mostly diagnosed prenatally, as opposed to EH shunts, which are commonly diagnosed later in life and are more likely to be symptomatic [1,8,13].

### 4.2. Prenatal US

With the improvement of US techniques, up to 42% of all types of CPSS are now diagnosed prenatally and these numbers are increasing [5]. Other congenital malformations are identified in as many as 65% of patients diagnosed prenatally, the most frequent being congenital cardiac anomalies [29,30].

### 4.3. Positive Neonatal Screening for Galactosemia

Positive neonatal screening for galactosemia is a well described clinical presentation reported in 13 to 30% of CPSS [1,12,26,31,32]. In this context, hypergalactosemia is a false positive, thought to result from ingested milk and absence of liver first pass, despite normal galactose-1-phosphate uridylyltransferase 1 activity [1,26].

### 4.4. Neonatal Cholestasis

Neonatal cholestasis is another frequent presentation of CPSS and has been described in up to 32% of patients in different cohorts [5,13,33]. The underlying pathophysiology is unclear but may be related to portal deprivation. Differentiating cholestasis as the cause or consequence of CPSS is difficult, as increased intrahepatic resistance due to cholestatic liver disease may divert portal flow through a shunt [3]. In this regard, identifying a CPSS does not obviate the need for a full cholestasis work-up (Key message Box).

### 4.5. Hepatic Manifestations

CPSS can result in the development of a wide range of benign and malignant liver nodules, including nodular regenerative hyperplasia, FNH, adenomas, hemangiomas, hepatoblastomas, hepatocellular carcinomas and sarcomas [1,5,15,16,17]. It has been reported that 25% of cases of CPSS are associated with the development of benign hepatic tumors and 4% with malignant ones [12]. In a review of 265 children, one quarter of patients developed nodules, of these 60% presented with more than one nodule, and 10% were malignant [1]. In this same review [1], nodules were present in patients with IH or EH shunts, but only EH shunts were associated with malignancy. As for benign nodules, many seem to disappear after shunt closure, although this is difficult to quantify owing to loss to follow-up. These lesions are thought to arise due to inadequate delivery of growth factors to the liver by lack of portal venous flow and the compensatory hepatic arterial buffer response [34,35]. In the adult multicenter series, nodules were present in >40% of cases at diagnosis and a total of 8/66 patients developed hepatocellular carcinoma [9].

Consistent with the role of portal flow contributing to liver health and growth, hypoplastic left liver lobes have been described in patients with PDV [36]. Reversible hepatic steatosis has also been reported in patients with PDV as a probable consequence of lack of hepatic first-pass [37].

Both hyperammonemia and elevated bile acids are common laboratory abnormalities found in patients with CPSS. Hyperammonemia has been described in 79% of children in a review of a large number of children with CPSS (123/156) [1], with plasma values ranging from 1.1 to 10 times normal levels [1], and is thought to contribute to neurological manifestations, including hepatic encephalopathy [3,4,5,12,33,38] (Key message Box). Elevated serum bile acids, described in 97% of children in the same cohort (76/78 children) [1], reflect portosystemic bypass after intestinal absorption [1,5]. Both are useful markers to monitor the effectiveness of shunt closure, as values decrease and most commonly normalize days after restoration of portal flow [1,3,4,5,9,33,39].

### 4.6. Cardiopulmonary Manifestations

Cardiopulmonary manifestations are among the most frequent—15 to 26% [1,5,10]—and most serious in CPSS. These include HPS, PAH and high-output cardiac failure. Both HPS and PAH may be more common in children with heterotaxia and/or polysplenia [10,40].

HPS is defined by the following triad: liver disease with or without portal hypertension, arterial oxygenation defect with or without hypoxemia and intrapulmonary vascular dilatations without associated cardiovascular disease [41]. It has been reported in 18% of patients with CPSS, led to diagnosis in 11% of cases and was observed in all types of CPSS [1]. Hypotheses regarding the pathophysiology behind the development of HPS in CPSS, as in other liver diseases, include microvascular alterations within the pulmonary arterial circulation linked partly to excess pulmonary production of vasodilators, such as nitric oxide and abnormal angiogenesis secondary to splanchnic-produced vascular endothelial growth factors, and other mechanisms [42].

PAH, defined as mPAP > 20 mmHg measured by right heart catheterization with pulmonary vascular resistance ≥ 3 wood units and pulmonary arterial wedge pressure ≤ 15 mmHg [43], has been reported in 11% of patients, led to the diagnosis in 7% of cases and was observed in all types of CPSS [1]. The pathophysiology of PAH in the setting of CPSS has been also linked to an imbalance between pulmonary vasodilators and vasonconstrictors and is considered a portopulmonary hypertension, although portal hypertension is not a feature of CPSS. In addition, there may be increased oxidative stress due to an imbalance between vascular inflammatory and anti-inflammatory mediators [44]. There are reports of HPS and PAH in the same patient either concurrently or sequentially, but this is a very rare occurrence [5]. Patients with moderate to severe PAH (mPAP ≥ 30 mmHg and >40 mmHg, respectively [43]) with clinical symptoms seem to benefit from early and aggressive medical treatment before shunt closure, enabling improvement of their hemodynamic profile [18].

Finally, cardiac failure has been described in CPSS patients due to high cardiac output [5,10], predominantly in fetuses and neonates, and impacted 16% out of 168 patients in a review [10]. Heart failure, with or without association to a congenital cardiac malformation, was the main cardiac symptom in both the prenatal and neonatal period, in contrast with HPS and PAH that were predominant after the first month of life [10]. Regardless of their association with a congenital cardiac malformation, these neonates displayed favorable outcomes under medical therapy suggesting that depending on the anatomy of the CPSS, supportive management may allow for shunt closure to be deferred until the patient is older and the technical feasibility is easier [10].

### 4.7. Neurocognitive Manifestations

Neurological abnormalities are among the most common and severe manifestations of CPSS, with a reported 29–35% of cases with either IH or EH shunts [1,9,12]. These abnormalities arise from any type of CPSS [1]. They range from mild cognitive deficits [25] to unexplained mental retardation [12,26], from attention–hyperactivity disorders and behavioral problems [1,26] to clear signs of encephalopathy [25]. There are also reports of post-prandial loss of consciousness or lethargy [1,37,45], seizures [1], Parkinson-like syndromes [46] or hepatic myelopathy [47,48]. Relevant findings include hyperammonemia [1,5,9,12,33,37,49], which has been linked to the ratio of portosystemic bypass [1], although plasma ammonia is an unreliable measure of encephalopathy in the absence of acute liver failure [50,51]. In addition, the hallmark sign of hepatic encephalopathy, hyperintense T1 signal of the globus pallidus on MRI, has been repeatedly reported [1,3,9,33,52,53].

### 4.8. Syndromic Associations

CPSS are associated with Down’s syndrome, polysplenia and heterotaxia [1,3,12,54] as well as other syndromes [11] (Table 2), which have been reviewed in detail elsewhere [12]. Thus, in case of visceral or cardiac malformations, it is recommended that CPSS be sought using liver Doppler US.

## 5. Other Symptoms and Signs of CPSS

Table 3 summarizes clinical and biological symptoms and signs of CPSS, which often arise together and should therefore prompt the search for a unifying etiology. The main clinical manifestations of CPSS are hepatic, cardiovascular and neurological [12], but CPSS can mimic other systemic conditions in an estimated 30% of all cases [5]. In children aged 1 month or more, 73% of CPSS were diagnosed while investigating signs or symptoms, such as neonatal cholestasis, hyperammonemia, liver tumors, HPS, PAH or encephalopathy [1]. Other symptoms and signs of CPSS include endocrine, gastrointestinal, hematological, immunological, cutaneous and renal manifestations. These symptoms and signs are outlined below.

### 5.1. Endocrine

Manifestations of CPSS highlight the key endocrine role of the liver. While intrauterine growth restriction is a common feature of patients with CPSS [1,29,62,63,64,65] and syndromic patients with CPSS may present with short stature [65], tall stature has also been described by centers following patients with CPSS [3]. Incidentally, tall stature and/or improvement of growth parameters was noted in the setting of surgical shunts [20,21]. Experts hypothesize that overgrowth in the setting of CPSS may be linked to abnormal growth hormone metabolism related to absent hepatic first pass and degradation, with a consequent altered secretion of insulin-like growth factor 1 [66,67] resulting in decreased negative feedback to the pituitary gland secreting growth hormone [68,69]. This is an area that needs further characterization.

The pathophysiology of hyperinsulinemia in CPSS is partly similar to that in cirrhosis and is linked to several mechanisms: absence of hepatic insulin metabolism after its secretion into the mesenteric and portal system [22,70]; excessive insulin secretion in response to peripheral hyperglycemia due to absence of hepatic first pass [19,71]; and insulin resistance due to the negative-feedback of hyperinsulinemia on insulin receptor-binding proteins [71]. Other endocrine manifestations of CPSS are likely secondary to hyperinsulinemia. One such example is hypoglycemia [1,19], particularly in the neonatal period [13,33]. Another is hyperandrogenism secondary to insulin-dependent ovarian and adrenal androgen production leading to amenorrhea [22]. Hyperandrogenism with precocious puberty may also occur in the setting of CPSS owing to proportionately decreased hepatic sulfation of dehydroepiandrosterone to the less active dehydroepiandrosterone sulfate, therefore leading to more potent circulating androgens [19]. Hyperandrogenism by itself and over time leads to accelerated somatic maturation and bone age, which therefore causes central precocious puberty and early acceleration of growth velocity [72]. It is unclear at the present time whether accelerated linear growth is due to growth hormone metabolism or hyperandrogenism.

Thyroid dysfunction has also been reported anecdotally in patients with CPSS, pointing to yet another endocrine role of the liver. Disruption of peripheral thyroid signaling and function in patients with CPSS is probably multifactorial. First, thyroid-binding protein (TBG) synthesis as well as the production of other thyroid-distributing proteins are at least in part dependent on T4 [73]. Therefore, in the absence of T4 hepatic first pass, TBG synthesis may be diminished, impacting delivery to peripheral tissue. Next, TBG synthesis has also been shown to be dependent on estrogen signaling [74]. Therefore, incomplete hepatic first pass likely contributes to aberrant peripheral thyroid signals as well as other endocrine abnormalities.

### 5.2. Gastrointestinal

Gastrointestinal manifestations of CPSS include abdominal pain, pancreatitis [55], rectal bleeding or protein-losing gastropathy [56]. Abdominal pain was a feature of 15% of patients with EH CPSS in an international observational study of 66 adult patients [9] and may be linked to spontaneous bleeding of hepatic tumors, although rare [57,58]. Rectal bleeding in CPSS is paradoxically unlinked to portal hypertension, but rather thought to occur owing to abnormal splanchnic venous drainage resulting in vascular congestion and relative intestinal mucosal ischemia and the consequences thereof [59,60,61]. Gastrointestinal symptoms being very common pediatric complaints, we recommend performing a liver Doppler US if these symptoms remain unexplained or present in a child with unclear multisystemic involvement.

### 5.3. Hematological/Immunological

Hematological and immunological manifestations are anecdotally reported in CPSS patients and are linked to the reticuloendothelial role of the liver for the latter [3]. Animal studies have demonstrated changes in levels of clotting factors and anticoagulants in the presence of CPSS [81]. In humans, cases are scarce, and include one patient with coagulation abnormalities on pre-operative work-up [33], a single report of vaginal bleeding linked to a meso-iliac fistula [56] and a retrospective report of 19 cases with mild coagulopathy [82]. Decreased factors V, VII-X and prolonged prothrombin time due to low grade consumption with or without thrombocytopenia have been suggested by experts as the causes of coagulopathy [3]. Low grade cholestasis owing to elevated serum bile acids secondary to impaired enterohepatic circulation may also contribute to a reduction in vitamin-K dependent factors.

Finally, as the liver’s reticuloendothelial function is bypassed in CPSS, patients are theoretically at risk of deep tissue infections. Central nervous system abscesses have been described in children with CPSS associated to intrapulmonary shunts [83,84,85,86].

### 5.4. Cutaneous Angiomas

Cutaneous angiomas were present in 5% of cases in a review of 265 patients with CPSS [5]. Although the relationship between CPSS and angiomas is unclear, this clinical finding should suggest CPSS, certainly when combined with other systemic complaints, and prompt a liver Doppler US.

### 5.5. Renal

Renal manifestations of CPSS have been occasionally reported, akin to what has been described in the setting of transjugular intrahepatic shunts [75]. Membranoproliferative glomerulopathy is hypothesized to result from an increase in circulating immune complexes formed in the portal circulation and not cleared by the liver’s reticuloendothelial system [76], as has been seen in patients with both congenital and surgically created portosystemic shunts, even in the absence of cirrhosis [23,77,78]. Disease severity can extend from mild proteinuria or hematuria to advanced nephrotic syndrome and glomerulopathy with full immunohistochemical patterns [23,77,79,80].

## 6. Approach for the Clinician

### 6.1. Diagnostic Approach

CPSS may masquerade as several pediatric disorders, making their diagnosis difficult. They should especially be sought when patients present with concurrent or sequential complaints of unclear etiology (Key message Box). Table 2 summarizes proposed indications for liver Doppler US in various systemic conditions. CPSS diagnosed prenatally always warrant a postnatal Doppler US, as the circulatory changes after birth significantly alter findings [5]. In all age groups, liver Doppler US is recommended as a screening and monitoring tool (Key message Box). Angio-CT scan is used to confirm CPSS presence, location and anatomical type, as well as the presence of liver nodules [5]. In confirmed cases of CPSS, identification of systemic manifestations is essential in outlining management and surveillance of the shunt, as covered in the preceding section. Imaging can be used to measure shunt size. Per-rectal scintigraphy may provide indirect quantification of the shunt ratio (also called shunt fraction or shunt index), i.e., the degree of flow traversing the shunt [1,12,87,88]. How this ratio is related to clinical signs and symptoms is unclear and still a matter of debate among experts.

### 6.2. Management: CPSS Closure and Management of Systemic Manifestations

A detailed description of the management of CPSS is beyond the scope of the present review and the reader is referred to other recent publications [1,3,4,5]. Nonetheless, clinicians will need several elements to guide their patients in the meanders of their sometimes complex, multidisciplinary management. Essentially, the goal of CPSS management is either to halt or reverse systemic manifestations and/or prevent their development in patients who have reached an age beyond which spontaneous closure can still be expected [3,4]. IH CPSS are more likely than EH shunts to close spontaneously (47% versus 4% of EH shunts), particularly when diagnosed before the age of 2 years [8,13,29,89]. However, there are no established predictors of spontaneous closure [13,89].

In this regard, expert opinion recommends that patients with a prenatal diagnosis of CPSS should be monitored at regular intervals (for example, at 1-3-6-12 months of age or according to local resources), both clinically and by imaging until first year of life, and less frequently thereafter. It is recommended that CPSS identified incidentally after birth be investigated with the basic work-up suggested in Table 4. Referral to a specialized center is advised for CPSS diagnosed incidentally after the age of 2 years or for symptomatic CPSS at any age. Indeed, current practice is to recommend closure in most CPSS that have not spontaneously closed after 2 years of age, although timing is still a matter of debate and type of closure is best evaluated in centers of expertise [1,3,5,8,13].

Other than neonatal cholestasis that may resolve spontaneously [5], benefits of closure include reversal of the main manifestations of CPSS. First, restoration of portal flow and first hepatic pass can be expected [4,9]. Next, regression of benign liver nodules is probable over time [1,4,9]. Third, resolution of HPS and hypoxemia can be expected, although close follow-up is required as persistence of pulmonary arteriovenous communications or development of PAH have been described [9,33,90,91]. Fourth, stabilization or improvement of PAH can be observed, although progression has been described [1,4,9,10,18]. Finally, regression of neurocognitive signs and symptoms, including encephalopathy, hyperammonemia and MRI findings, have been reported in the majority of children [1,5,9,33]. Renal and endocrine outcomes after CPSS closure have been less documented.

Endovascular closure is the preferred method in centers with the necessary expertise, although some types of CPSS warrant surgical closure. Liver transplant, used historically as a treatment method, is now a matter of debate. While some centers favor liver transplant in some indications [92,93], it is mainly performed in case of failure of shunt closure, in the presence of malignant nodules, or if severe life-threatening complications, such as portal hypertension or worsening pulmonary hypertension, occur after shunt closure [3,5,57,93,94].

The main complications following closure, other than the inherent risks of anesthesia especially in the setting of PAH, are mesenteric or portal thrombosis [5], plug migration or new-onset portal hypertension [3]. In contrast with acquired shunts in the setting of portal vein occlusion or liver cirrhosis, CPSS typically do not present with portal hypertension [12]. Therefore, new-onset portal hypertension following CPSS closure may be linked to occlusion anywhere in the porto–mesenteric axis, including cavernous transformation of the portal vein. It may also be due to underlying liver fibrosis, highlighting the importance of liver biopsy and occlusion test prior to shunt closure (Table 2) [5].

Pre-emptive closure is controversial. Nonetheless, at the present time, it is generally accepted that in most situations, the benefits likely outweigh the risks for several reasons, by protecting patients from such life-threatening complications as PAH. That said, it is still unknown who will develop complications, something which would help inform and personalize management decisions [1,5]. Finally, it is important to consider that the young intra-hepatic portal tree may be more adaptable to restored portal flow than that of an older subject.

Although detection of CPSS is on the rise through increased awareness and improved imaging, much is still unknown. It is unclear which patients are at risk of developing complications and consequently who will most benefit from closure. Surveillance of liver nodules is another challenge that requires further study. The aim of the International Registry of Congenital Porto-Systemic Shunts (http://ircpss.com/IRCPSS.html, accessed on 20 November 2021) is to better identify patients at risk of developing complications and offer standardized care.

## 7. Conclusions

These three cases illustrate some of the presentations of CPSS in children: liver nodules, neurocognitive difficulties, PAH and possibly tall stature. Liver Doppler US is the key to diagnosis. Considering CPSS in the diagnostic work-up of a patient with a complex clinical picture of unexplained endocrine, liver, gastro-intestinal, cardiovascular, hematological, renal or neurocognitive disorder is important, as prompt referral to a specialized center may significantly impact patient outcome.


**Key Message Box**
Suspect CPSS in patients with a constellation of seemingly unrelated symptoms.Liver Doppler US is the first essential step for diagnosis.Identification of a CPSS does not obviate the need for a full cholestasis work-up.Hyperammonemia of unexplained etiology in a neonate or child should prompt screening for CPSS. 

## Figures and Tables

**Figure 1 children-09-00243-f001:**
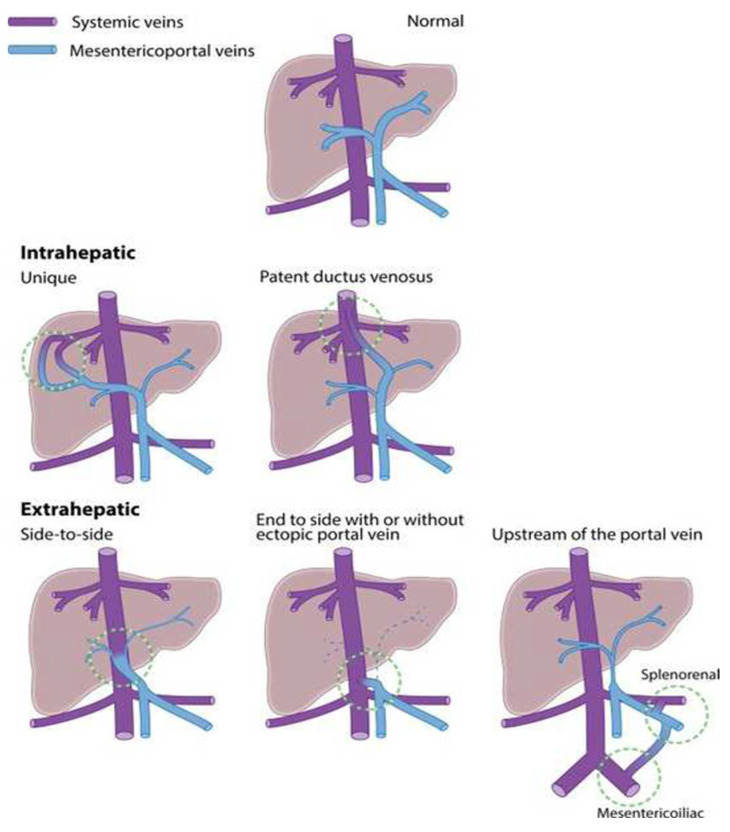
Anatomical forms of congenital portosystemic shunts. Reproduced from Guérin et al. Congenital portosystemic shunts: Vascular liver diseases: Position papers from the francophone network for vascular liver diseases, the French Association for the Study of the Liver (AFEF), and ERN-rare liver. Clin. Res. Hepatol. Gastroenterol. 452–459. Copyright © 2022 Elsevier Masson SAS. All rights reserved [5].

**Figure 2 children-09-00243-f002:**
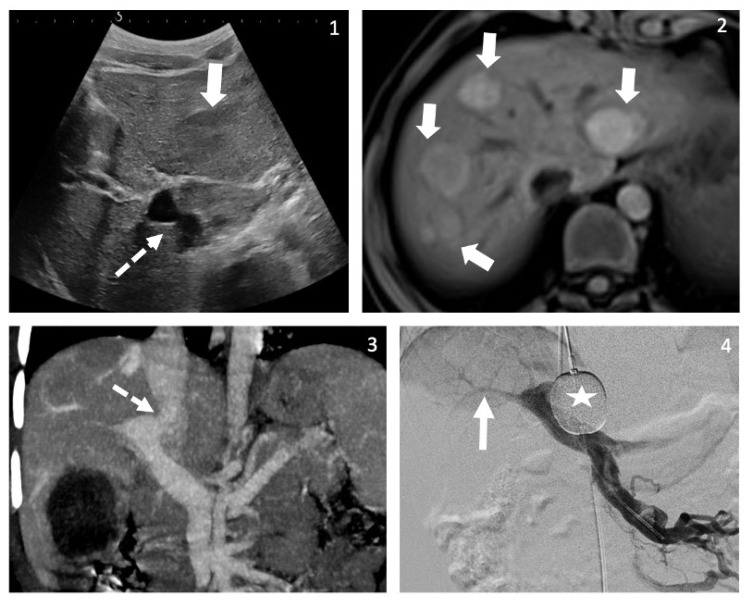
Case C. Aspect on imaging of the congenital porto-systemic shunt (CPSS) and the liver nodules. (**1**) Axial view on US B-mode at the level of the portal bifurcation shows the direct communication between the portal bifurcation and the inferior vena cava (IVC) (dotted arrow). There is a large liver tumor in the left hepatic lobe (large white arrow). (**2**) Axial view on MRI with contrast injection at the arterial phase shows the strong early enhancement of multiple hepatic nodules corresponding to hepatocellular proliferation. (**3**) Coronal view of an abdominal contrast enhanced computed tomography (CT) at the portal phase shows the direct communication between the portal bifurcation and the IVC. (**4**) Direct portal venography after occlusion of the CPSS with a balloon inflated in the IVC (star) shows hypoplastic right portal vein that was not visible on non-invasive imaging (plain arrow).

**Table 1 children-09-00243-t001:** Summary of three cases of congenital portosystemic shunts presenting with tall stature (>percentile 97). PAH = pulmonary arterial hypertension; US = ultrasound; FNH = focal nodular hyperplasia.

Patient	Cause for Referral	Age at First Symptoms/Signs (Years)	Age at Diagnosis (Years)	Cardiovascular Symptoms	Liver Nodules	Other Endocrine Symptoms
A	Suspected PAH	9	11	Dyspnea	Beta-catenin mutated adenomas	Insulin resistance with hyperinsulinism, acanthosis nigricans and hyperandrogenemia without menarche
B	Elevated aminotransferase levels	0	7	No	Multiple adenomas	Neonatal hypoglycemia
C	Liver nodules identified on US performed for abdominal pain	3	5	No	Beta-catenin mutated FNH-like nodules	No

**Table 2 children-09-00243-t002:** Indications for liver Doppler US.

Indications for a Liver Doppler Ultrasound [References]				
Syndromes Associated with CPSS [1,3,11,12,54]	Visceral Malformations	Cardiac Defects	Liver Malformations	Other
Caroli’s	Mesenteric defects	Ventricular septal defects	Biliary atresia:	Membranoproliferative glomerulonephritis
Goldhenhar	Duodenal atresia	Atrial septal defects	- Syndromic	Unexplained symptoms/signs in Table 3
Down’s	Ano-rectal malformations	Hypoplastic left heart	- Non syndromic	
Turner	Polyposis syndromes	Left isomerism	Masses:	
Leopard	Situs abnormalities	Valvular abnormalities	- Hepatoblastoma	
Rendu-Osler	Renal malformations		- Hepatocellular carcinoma	
Grazioli			- Other	
Noonan			Antenatal abnormal imaging:	
Cornelia de Lange			- Left lobe hypoplasia	
Holt-Oram				
Costello				
Wolf-Hirschhorn				
Neurofibromatosis				
Adams-Oliver				

**Table 3 children-09-00243-t003:** Clinical and biological symptoms and signs encountered in patients with congenital portosystemic shunts. Signs and symptoms not in order of frequency of presentation. ADHD = attention deficit hyperactivity disorder; LOC = loss of consciousness.

Clinical and Biological Symptoms & Signs Encountered in Patients with Congenital Portosystemic Shunts [References]						
Hepatic [1,3,4,5,9,12,15,16,17,33,34,35,36,37,38,39]	Gastro- Intestinal [9,55,56,57,58,59,60,61]	Cardio- Pulmonary [1,5,10,18,40,41,42,43,44]	Endocrine/Metabolic [1,3,13,19,20,21,22,29,33,62,63,64,65,66,67,68,69,70,71,72,73,74]	Renal [23,75,76,77,78,79,80]	Neurocognitive [1,5,9,12,25,26,33,37,45,46,47,48,49,50,51,52,53]	Other [3,5,33,56,81,82,83,84,85,86]
Abnormal hepatic vasculature on antenatal ultrasound	Abdominal pain	Hepatopulmonary syndrome	Hyperinsulinemic hypoglycemia	Proteinuria	Mild cognitive deficits	Brain abscesses (when associated with intrapulmonary shunts)
Tumors:	Gastrointestinal bleeding	Pulmonary artery hypertension	Hyperandrogenism	Hematuria	ADHD	Coagulation abnormalities
- Nodular regenerative hyperplasia		High-output cardiac failure	Precocious puberty		Post-prandial LOC	Cutaneous and visceral hemangioma
- Focal nodular hyperplasia			Amenorrhea		Parkinson-like	
Adenoma			Hypothyroidism		Hepatic myelopathy	
- Hepatoblastoma			Fetal growth retardation		Portosystemic encephalopathy	
- Hepatocellular carcinoma			Tall stature/overgrowth		Learning difficulties	
Hemangioma			Hyperammonemia		Unexplained mental retardation	
Hypoplastic left liver			Elevated serum bile acids			
Steatosis			Increased galactose on newborn screen			
Portal hypertension						

**Table 4 children-09-00243-t004:** Purpose of recommended basic work-up in suspected CPSS.

	Recommended Basic Work-Up when Suspecting a CPSS [3,52]
	**Thorough clinical examination**	**Thoraco-abdominal CT-angiography**
	Other malformations	Anatomy of CPSS
	Cutaneous hemangiomas	Hepato-pulmonary shunts
	**Laboratory**	**Myocardial contrast echocardiography**
	Elevated serum aminotransferases	Evidence of right-to-left shunting
Expected	Abnormal coagulation	If elevated right sided pressure: right heart catheterization
findings	Elevated fasting bile acids	
	Elevated fasting ammonia	
	**Abdominal Doppler ultrasound**	**Brain MRI**
	Liver masses	T1 hyperintensity in globus pallidus
	**Liver biopsy**	
	Underlying liver disease	
	Liver nodules

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
