# Peer review of "Presentation of Congenital Portosystemic Shunts in Children"

_children, 2022, doi:10.3390/children9020243_

Round 1

Reviewer 1 Report

The authors present a review article on Presentation of Congenital Portosystemic Shunts in Children. CPSS are a complicated topic and to present the topic in a manner that is clear and easy to understand for the readers is challenging. There are many reports in the literature that add to the confusion in recognition and management of patients with this condition. In this regard, a quality systematic review would be a welcome addition to the scientific literature.

The advantage of this manuscript is that it approaches the topic systematically from a broad perspective. To keep the manuscript to a manageable length, the treatment for this condition is only barely discussed, which is understandable.

There are however some parts where the manuscript could be improved. Some specific comments by section are listed below:

Abstract

Page 1, line 24: Please add word “venous”, for the sentence to read: communications between the portal venous system and the systemic VENOUS circulation

Page 1, line 26: tall stature is a clinical sign that was present in the three cases you highlighted, however, as this was not a study to determine the connection between tall stature and CPSS, and it has not been previously generally associated with this condition, I would not suggest to point it out specifically here – also, in conclusion, you don't propose any different diagnostic approach to children with tall stature in specific, so it is not clear why it should be highlighted here

Page 1, line 42: in the cases described above, CTA seems to be the key to diagnosis, here it is pointed out that Doppler ultrasound is the key. The readers may get confused.

Introduction

Page 2, line 52: Please add word “venous”, for the sentence to read: communications between the portal venous system and the systemic VENOUS circulation

Page 2, line 62: »extrahepatic (EH) shunts rarely close spontaneously« - this statement would require a citation. One of the articles cited at the end of the sentence specifically states that extrahepatic shunts don't close spontaneously. Thus, a study (or a case report) proving otherwise at this point of sentence would be required.

Page 2, line 63: »while intrahepatic (IH) shunts seem more prone to do so in early life« All the cited references here are review articles. For this statement a good reference could be this article, with follow up of intrahepatic shunt that spontaneously closed: Plut D, Gorjanc T. A case of a newborn with an intrahepatic congenital portosystemic venous shunt… Acta Radiol Open. 2019; DOI: 10.1177/2058460119854173

Page 2, line 75: Contrary to tall stature, there are reports of small growth in children with CPSS as well. Reference: Passanisi et al. Congenital Portosystemic Shunt: Our Experience. Case Reports in Pediatrics 2015, DOI: 10.1155/2015/691618

Page 3, line 83: As previously mentioned, I don’t think tall stature should be the focus of this review article. It’s a review article after all. A different kind of paper/study would be required to determine the association between CPSS and tall stature – the tall stature in these cases could be related to liver adenomas, genetic syndrome or other abnormality and may not be specifically linked to CPSS.

Case presentation

Page 4, lines 115-133: Case 2 would require further clarification. At first, the absence of the main portal vein is described and direct communication between spleno-mesenteric confluence to the IVC is described. Later, this confluence seemed to be partially banded – were there small veins leading to the liver seen at surgery? A small portal vein? Later shunt thrombosis is described and portal vein thrombosis as well (however, previously it is reported that the portal vein was absent) – please elaborate on this case.

Typical clinical presentations of CPSS

Page 4, line 153: In this first general paragraph about clinical presentations, I would suggest adding a few sentences that explain that the size of the shunt is important for the presence of symptoms. Therefore, it is not always important if the shunt is EH or IH, but the % of blood that bypasses the liver determines the symptoms. Your cited reference #1 (Paganelli) or this one “Yoon HK, Choo SW, Do YS, et al. Congenital intrahepatic portosystemic shunts in the neonate: coil embolization via the umbilical vein. J Vasc Interv Radiol 1998;9:509–511« are a good reference.

Page 5, line 186: »nodules were present in patients with IH or 185 EH shunts, but only EH shunts were associated with malignancy« How did this correlate to the size of the shunt? (% of diverted blood)

Page 6, line 250: »although plasma ammonia is not a good measure of portosystemic bypass or of encephalopathy« this statement requires a citation.

Page 6, lines 257: »Thus, in case of visceral or cardiac malformations, it is recommended that CPSS be sought using Doppler US«. In the cases you presented, as well as many cases in the literature, we can see that these shunts are often missed on Doppler US – do you think CTA should be performed or maybe a better question, »At what time should CTA be performed? « Discuss on this in the article please. Expand the paragraph on diagnostic approach on page 9.

Page 10, line 391: Liver transplant is still warranted in many types I EH shunts (if occlusion test results are poor).

Page 10, line 407: As mentioned above, the size of the shunt (% of diverted blood) is an important matter.

Conclusion

Page 10, lines 417-422: »Abdominal Doppler US is the key to diagnosis. « Although I agree with this statement, the three presented cases don't demonstrate this well. Expand on diagnostic approach in the manuscript and in the conclusion.

Figures and Tables

Figure: Appropriate. Some radiologic imaging of the shunt would be welcome. At least a couple basic types (EH, IH; US and CTA).

Tables:

Table 3 is listed in the text before Table 2.

Table 2:

I would suggest changing the title of the table to »Clinical and Biological Symptoms & Signs Encountered in Patients with Congenital Portosystemic Shunts (references)». As some of these are described in case reports, direct attribution of the sign/symptom to CPSS is difficult. A detailed study would be required to describe associations between CPSS and all these signs & symptoms.

Table 3: What does »hard indication« mean? What exactly is meant as »abdominal Doppler-ultrasound« (this actually goes for the manuscript overall)? Is this an evaluation of blood flows of liver veins? (hepatic artery, hepatic veins, portal veins?) – in that case, »a liver Doppler-US« would be a better term. There are a lot of veins and arteries in the abdomen.

Table 4: Again, the same issue with »Abdominal Doppler ultrasound« as above. Liver masses are detected on abdominal ultrasound – most of the time, Doppler technique is not needed to detect a liver mass. Doppler may help in characterization of masses, but even more in evaluation (and detection) of the CPSS. Although, most of the time you don’t need Doppler technique to detect it.

Additionally, elaborate on T1 hyperintensity of MRI – be more specific about this finding (area). T1 hyperintensity in general can be a sign of many diseases. In this case, it is specific in certain areas.

Author Response

Dear Reviewer,

Thank you for your time and key suggestions. Please find below in blue answers to your specific comments, highlighted in yellow in the manuscript.

Comments and Suggestions for Authors

The authors present a review article on Presentation of Congenital Portosystemic Shunts in Children. CPSS are a complicated topic and to present the topic in a manner that is clear and easy to understand for the readers is challenging. There are many reports in the literature that add to the confusion in recognition and management of patients with this condition. In this regard, a quality systematic review would be a welcome addition to the scientific literature.

The advantage of this manuscript is that it approaches the topic systematically from a broad perspective. To keep the manuscript to a manageable length, the treatment for this condition is only barely discussed, which is understandable. There are however some parts where the manuscript could be improved. Some specific comments by section are listed below:

Abstract

Page 1, line 24: Please add word “venous”, for the sentence to read: communications between the portal venous system and the systemic VENOUS circulation。

Added.

Page 1, line 26: tall stature is a clinical sign that was present in the three cases you highlighted, however, as this was not a study to determine the connection between tall stature and CPSS, and it has not been previously generally associated with this condition, I would not suggest to point it out specifically here – also, in conclusion, you don't propose any different diagnostic approach to children with tall stature in specific, so it is not clear why it should be highlighted here.

Expert opinion is that tall stature is a common feature of CPSS, but indeed, because of the dearth of data, this feature has been removed from the abstract and will be further discussed in the text.

Page 1, line 42: in the cases described above, CTA seems to be the key to diagnosis, here it is pointed out that Doppler ultrasound is the key. The readers may get confused.

We specified in the abstract that Doppler US was used as the first imaging technique and CTA as confirmatory imaging in our 3 patients.

Introduction Page 2, line 52: Please add word “venous”, for the sentence to read: communications between the portal venous system and the systemic VENOUS circulation.

Added.

Page 2, line 62: »extrahepatic (EH) shunts rarely close spontaneously« - this statement would require a citation. One of the articles cited at the end of the sentence specifically states that extrahepatic shunts don't close spontaneously. Thus, a study (or a case report) proving otherwise at this point of sentence would be required.

References added.

Page 2, line 63: »while intrahepatic (IH) shunts seem more prone to do so in early life« All the cited references here are review articles. For this statement a good reference could be this article, with follow up of intrahepatic shunt that spontaneously closed: Plut D, Gorjanc T. A case of a newborn with an intrahepatic congenital portosystemic venous shunt… Acta Radiol Open. 2019; DOI: 10.1177/2058460119854173

Thank you for this suggestion. Reference added.

Page 2, line 75: Contrary to tall stature, there are reports of small growth in children with CPSS as well. Reference: Passanisi et al. Congenital Portosystemic Shunt: Our Experience. Case Reports in Pediatrics 2015, DOI: 10.1155/2015/691618

In the proposed study, one of the patients is described as having a height <P90, the other has Down syndrome, which may explain the small stature. Nevertheless we further expanded on the possibilities of small stature in children with CPSS in the Endocrine section (page 9, line 299).>< P90, the other has Down syndrome, which may explain the small stature. Nevertheless we further expanded on the possibilities of small stature in children with CPSS in the Endocrine section (page 9, line 299).

Page 3, line 83: As previously mentioned, I don’t think tall stature should be the focus of this review article. It’s a review article after all. A different kind of paper/study would be required to determine the association between CPSS and tall stature – the tall stature in these cases could be related to liver adenomas, genetic syndrome or other abnormality and may not be specifically linked to CPSS.

While adenomas have been described in overgrowth syndromes, we are not aware of evidence linking tall stature to adenomas in the setting of CPSS. We highlight in our endocrine section and conclusion that further studies in this area are warranted. While this is a review article, it is also a case series of which patients did present with tall stature. Nevertheless, tall stature is a striking finding of the cases presented herein and therefore we left it in the text.

Case presentation

Page 4, lines 115-133: Case 2 would require further clarification. At first, the absence of the main portal vein is described and direct communication between spleno-mesenteric confluence to the IVC is described. Later, this confluence seemed to be partially banded – were there small veins leading to the liver seen at surgery? A small portal vein? Later shunt thrombosis is described and portal vein thrombosis as well (however, previously it is reported that the portal vein was absent) – please elaborate on this case.

The shunt and right intrahepatic portal vein were thrombosed, which is now specified in the text (page 5, line 137).

Typical clinical presentations of CPSS

Page 4, line 153: In this first general paragraph about clinical presentations, I would suggest adding a few sentences that explain that the size of the shunt is important for the presence of symptoms. Therefore, it is not always important if the shunt is EH or IH, but the % of blood that bypasses the liver determines the symptoms. Your cited reference #1 (Paganelli) or this one “Yoon HK, Choo SW, Do YS, et al. Congenital intrahepatic portosystemic shunts in the neonate: coil embolization via the umbilical vein. J Vasc Interv Radiol 1998;9:509–511« are a good reference.

In Sokollik et al (2013), shunt size was calculated for 71 patients, either by direct shunt measurement on imaging or by perrectal scintigraphy: although EH were found to be larger than IH shunts (no statistical calculation), 4 patients had no linear correlation between diameter and shunt fraction, indicating that caution is required when drawing conclusions between shunt size and clinical outcomes.

In Cho et al (2014), shunt diameter increased in patients with both IH or EH shunts over time, however, shunt index, calculated by per-rectal portal scintigraphy and reflective of shunt severity, decreased with time in EH shunts.

A thorough literature search showed no clear association between shunt diameter and symptoms, although some degree of normal portal flow seems to be protective. We respectfully suggest that this is beyond the scope of a review for a general reader, especially as the evidence is lacking.

Page 5, line 186: »nodules were present in patients with IH or 185 EH shunts, but only EH shunts were associated with malignancy« How did this correlate to the size of the shunt? (% of diverted blood)

Unfortunately, shunt size was not analyzed in these cases.

Most centers do not practice per-rectal portal scintigraphy or measure shunt size on crosssectional imaging.

Page 6, line 250: »although plasma ammonia is not a good measure of portosystemic bypass or of encephalopathy« this statement requires a citation.

In Bernard et al, the shunt ratio is significantly related to the level of ammonemia (p=0.0009; Student’s test) and to the presence of clinical signs of encephalopathy (p=0.0062; Fisher’s exact test). However, in Kundra et al, plasma ammonia was not a good measure of encephalopathy, in the absence of acute liver failure and in Srivastava et al (Figure 5, ROC curve), plasma ammonia was not a reliable marker to differentiate children without minimal hepatic encephalopathy (MHE) from children with MHE.

Page 6, lines 257: »Thus, in case of visceral or cardiac malformations, it is recommended that CPSS be sought using Doppler US«. In the cases you presented, as well as many cases in the literature, we can see that these shunts are often missed on Doppler US – do you think CTA should be performed or maybe a better question, »At what time should CTA be performed? « Discuss on this in the article please. Expand the paragraph on diagnostic approach on page 9.

Our review being aimed at general pediatricians, we suggest performing a liver Doppler US as a screening tool. Referral to a specialized center is recommended in the following cases: a shunt upstream to portal bifurcation, a symptomatic shunt, or if the shunt persist beyond the age of 2 years. Referral centers can then complete the recommended basic work-up described in Table 4, which includes an angio-CT scan to further characterize shunt anatomy (paragraph on management).

Page 10, line 391: Liver transplant is still warranted in many types I EH shunts (if occlusion test results are poor).

The two-step surgical approach (Franchi-Abella et al, 2010) has allowed closure of shunts despite poor occlusion test results (Ushida et al 2021, Blanc et al 2014). Surgical methods are not discussed in present review and the reader is referred to numerous recent publications.

Page 10, line 407: As mentioned above, the size of the shunt (% of diverted blood) is an important matter.

Cf previous answer to first comment in same section.

Conclusion

Page 10, lines 417-422: »Abdominal Doppler US is the key to diagnosis. « Although I agree with this statement, the three presented cases don't demonstrate this well. Expand on diagnostic approach in the manuscript and in the conclusion.

Cf previous answer to third comment in “Typical clinical presentations of CPSS”

Figures and Tables

Figure: Appropriate. Some radiologic imaging of the shunt would be welcome. At least a couple basic types (EH, IH; US and CTA).

By mean of example, images from case C have been added (figure 2). Readers may refer to figure 1 for images of intra-hepatic shunts.

Tables:

Table 3 is listed in the text before Table 2.

Thank you for catching this! Table numbers were modified accordingly.

Table 2:

I would suggest changing the title of the table to »Clinical and Biological Symptoms & Signs Encountered in Patients with Congenital Portosystemic Shunts (references)». As some of these are described in case reports, direct attribution of the sign/symptom to CPSS is difficult. A detailed study would be required to describe associations between CPSS and all these signs & symptoms.

Title has been changed.

Table 3: What does »hard indication« mean? What exactly is meant as »abdominal Dopplerultrasound« (this actually goes for the manuscript overall)? Is this an evaluation of blood flows of liver veins? (hepatic artery, hepatic veins, portal veins?) – in that case, »a liver DopplerUS« would be a better term. There are a lot of veins and arteries in the abdomen.

Abdominal Doppler US was changed to hepatic Doppler US in manuscript. “Hard indications” was erased for more clarity.

Table 4: Again, the same issue with »Abdominal Doppler ultrasound« as above. Liver masses are detected on abdominal ultrasound – most of the time, Doppler technique is not needed to detect a liver mass. Doppler may help in characterization of masses, but even more in evaluation (and detection) of the CPSS. Although, most of the time you don’t need Doppler technique to detect it.

Abdominal Doppler US was changed to liver Doppler US in manuscript.

Additionally, elaborate on T1 hyperintensity of MRI – be more specific about this finding (area). T1 hyperintensity in general can be a sign of many diseases. In this case, it is specific in certain areas.

T1 hyperintensity in globus pallidus was added.

Reviewer 2 Report

This paper reviews the important clinical manifestations of CPSS through three clinical cases. 1. Since this is a review article, it is unnecessary to introduce three clinical cases. The clinical manifestations of these three cases are not common, and the clinical manifestations of CPSS are various. These three cases alone can not cover the clinical manifestations of CPSS. Therefore, the three cases should be improved to write a case report, or deleting these three clinical cases and writing a real review articles. 2. I haven’t meet the CPSS patients with tall stature in our more than 20 cases with CPSS (intra- and extra-hepatic). I don't know whether there is some ethnic difference in CPSS. 3. The three cases lack important data, such as diagnosed imaging evidence and the diameter of shunt vessels. 4.Table 1 should be improved: what kind of shunt is it? Intrahepatic or extrahepatic, location of shunt, Treatment method and prognosis. 5.Case B: What’s the duration of follow-up? What is the prognosis? Whether or not there is upper gastrointestinal bleeding caused by portal hypertension after surgery. In general, the thrombosis of the main portal vein will lead to cavernous transformation of the portal vein, which eventually needs to be treated by a surgical method. 6.Is the clinical manifestation related to the location of shunt? Some study found that the associated clinical symptoms are different owing to different analytical locations of the portosystemic shutdown (Zhang JS, Li L. surgical ligation of a portosystemic shutdown for the treatment of type II Abernethy malformation in 12 children. Journal of vascular surgery: Venus and lymphatic disorders, 2021,9 (2): 444-451.) 7.Part 5 "diagnostic approach" is obviously not comprehensive. Although ultrasound is the most easily available examination method, CT and angiography are still needed to make a clear diagnosis. Three cases in this study were clearly diagnosed by CT and angiography. Therefore, ultrasound is not a diagnostic method, it can only be used as a preliminary screening. 8.Because there are still some differences in the pathogenesis of intrahepatic and extrahepatic CPSS, and the most obvious difference is the spontaneous closure rate. There may be some differences in clinical manifestations between the two types. Please describe the different presentation of intrahepatic and extrahepatic CPSS.

Round 2

Reviewer 1 Report

Thank you for producing a revision of the manuscript. I believe the manuscript has been considerably improved. The added figures are a good addition for readers to visualize and understand the shunt, although images of all three presented cases would be even better in my opinion.

I have only two concerns to mention:

  1. Case B is still not completely clear to me. In this case, as I understand it, there was no main portal vein and the blood from mesenteric and splenic vein all diverted through the shunt to IVC. Clamping this shunt should therefore cause congestion in the bowel and spleen (poor perfusion is described). During surgery, partial banding was performed. After partial banding, thrombosis oh this shunt developed. Where did the blood flow? Was there a hypoplastic portal vein afterall that was visualized after the clamping?
  2. In the response to the reviewer letter, you provide some quality discussion about the shunt index (% of portal blood bypassing the liver), reflective of shunt severity. Although all the evidence about the clinical value of this shunt feature is not yet completely clear, and as you stated, many articles did not report this information, it may turn out to be an extremely important information in the management of CPSS. I would mention this in the manuscript, so readers could at least hear about it and researchers should be aware that this is an important feature of the shunt to measure and report (Doppler US can also provide at least approximation of this shunt severity).

Reviewer 2 Report

My final review comment and decision is “Reject”. The whole article is perfunctory to my comments.